# Clinical, Histological and Genetic Characterisation of a Disorder of Sexual Development in a Pygmy Goat

**DOI:** 10.3390/ani15070976

**Published:** 2025-03-28

**Authors:** Alberto Luque Castro, Melissa M. Marr, Emily L. Clark, Jacqueline Poldy, Lily Liu, Carola Daniel, Alexandra Malbon, Robert Kelly, Fraser Murdoch, Alastair Macrae, Neil Sargison

**Affiliations:** The Royal (Dick) School of Veterinary Studies and the Roslin Institute, University of Edinburgh, Easter Bush Campus, Roslin, Midlothian EH25 9RG, Scotland, UK; mmarr3@exseed.ed.ac.uk (M.M.M.); emily.clark@roslin.ed.ac.uk (E.L.C.); j.poldy@sms.ed.ac.uk (J.P.); liulily0518@163.com (L.L.); carola.daniel@ed.ac.uk (C.D.); alexandra.malbon@ed.ac.uk (A.M.); robert.kelly@roslin.ed.ac.uk (R.K.); fraser.murdoch@ed.ac.uk (F.M.); a.i.macrae@ed.ac.uk (A.M.); neil.sargison@ed.ac.uk (N.S.)

**Keywords:** disorder of sexual development, karyotype, pygmy goat, whole genome sequencing

## Abstract

This study presents a comprehensive investigation of a disorder of sexual development (DSD) in a pygmy goat, expanding current understanding beyond polled intersex syndrome. The individual goat presented with both male and female characteristics. Histological and genetic investigations revealed the presence of testicular tissue and implied an XY karyotype, resembling Swyer syndrome in humans. These findings contribute to the understanding of DSDs in goats, albeit further genetic investigations would be needed to determine the underlying cause.

## 1. Introduction

Historically, individuals with characteristics of both male and female phenotypes were described as intersex, hermaphrodites, pseudohermaphrodites, or sex reversals. Following a change in nomenclature regarding humans with such characteristics [1], these and a variety of other congenital presentations that arise from anomalous sexual differentiation are now classified together as disorders of sexual development (DSD) [2]. The entire spectrum of DSD can be categorised according to chromosomal, gonadal, and phenotypic presentation. Accordingly, chromosomal or genetic DSD results from anomalies in sex chromosomes and combination, whereas gonadal DSD describes normal XX or XY karyotype but discordance with the gonadal and/or external genital phenotype [3].

DSD is well documented in goats as they have a higher frequency of DSD than other domestic species (up to 10–15% in some breeds) [4]. These cases occur almost exclusively in polled (hornless) 60, XX individuals, a condition known as polled intersex syndrome (PIS) [5,6]. PIS is a heritable trait associated with an autosomal recessive expression, which exhibits a link to inheritance with the hornless locus, which has a dominant mode of inheritance. Individuals that are XX homozygous for the polled gene demonstrate a range of masculinisation, with testicular embryonic development in the absence of an *SRY* gene [5,6,7]. Rare examples of chimeric individuals with a XX/XY karyotype, such as freemartins, have been described [8]; nevertheless, only about 1% of heterosexual twin pregnancies in goats result in freemartins [9]. Swyer syndrome, a human condition describing a suite of presentations where an individual has a male XY karyotype but has female genitalia and a female or mixed phenotype [10], has been reported in a variety of domestic bovids, including cattle [11,12] buffalo [13], and sheep [14,15], but no unifying genetic basis has been found. This syndrome is also associated with an underdeveloped reproductive tract, elongated gonads (streak gonads), and, in some cases, clitoromegaly [10,16], as observed in the specimen of this study.

Although a variety of genetic, hormonal, and/or environmental possibilities exist to explain the development of a DSD, cases in livestock species rarely benefit from the comprehensive investigations made available for companion animals. Thus, full classification of individuals with ambiguous sexual features may be lacking. This work describes the clinical, histopathological, and genomic investigation of a phenotypically mixed female horned pygmy goat in order to characterise a DSD.

## 2. Materials and Methods

### 2.1. Animal Material

The affected individual was a 10-month-old assumed female pygmy goat (*Capra aegagrus hircus*). It was referred to the Farm Animal Hospital at the Royal (Dick) School of Veterinary Studies, University of Edinburgh, in January 2022 with a suspected diagnosis of a disorder of sexual development (DSD).

### 2.2. Diagnostic Imaging

Following clinical examination, transabdominal ultrasound (Mindray DP-30; IMV imaging, Strathclyde, Scotland) was undertaken using a 5.0–8.5 MHz micro-convex sector transducer, followed by computed tomography (CT) under general anaesthesia to evaluate the internal reproductive structures. CT was performed with the patient in sternal recumbency, using a 64-slice SOMATOM Definition AS (Siemens, Erlangen, Germany).

### 2.3. Surgical and Histological Examination

Exploratory laparotomy was performed under general anaesthesia, and the uterine and gonadal tissues were removed en bloc by standard surgical methods. The excised reproductive organs were fixed in 10% neutral buffered formalin, from which selected samples were paraffin-embedded and processed as 5-μm sections and stained with haematoxylin and eosin. Paired venous blood samples (heparin and EDTA tubes) were obtained prior to and two weeks after surgery for measurement of testosterone concentration and genetic analyses (stored frozen until analysis). A blood sample for cytological analysis to confirm the karyotype was not collected at the time, and as such, we used an approach based on whole-genome sequencing analysis as an alternative.

### 2.4. Samples, DNA Extraction and Sequencing

The suspected DSD goat sample was referred to as sample “DSD Goat”. DNA was extracted from peripheral blood by the commercial sequencing provider Neogen. Illumina short-read, paired-end (PE) libraries were prepared by Neogen and sequenced on the Illumina NovaSeq platform to a depth of 15 million reads per sample. Sequencing data from a control male and control female goat was used to facilitate comparative analysis. These two animals, “Male control” and “Female Control” (Appendix A), were mixed-breed dairy goats with whole genome sequencing available from a previous study [17]. The two control animals were sequenced to a higher depth of coverage than the patient (30X) DSD goat (15X), so these data were down-sampled to create alignments of comparable coverage. Our hypothesis was that the read coverage would act as a proxy for the karyotype and that an XY or intermediate karyotype could be identified by the proportion of reads mapping to each sex chromosome. This method has been used successfully to investigate ploidy in cell lines in livestock [18].

### 2.5. Illumina Read Mapping

Raw reads from all samples were quality trimmed and had remnant adapter sequences removed with Trim Galore, a Perl wrapper for Cutadapt v.1.18 [19] and FastQC v.0.12.1 [20]. Reads were mapped to two reference genomes: *Capra hircus* Saanen_v1 genome (GCA_015443085.1), derived from a European Swiss dairy goat breed, and the official goat reference genome ARS1.2 (GCF_001704415.2), derived from a San Clemente Island goat. The Saanen genome has an assembled Y chromosome, whereas the Y chromosome on the ARS1.2 reference assembly is currently on unplaced scaffolds, so the Saanen reference was used to examine this region. Reads were mapped using the Burrow-wheeler aligner v. 2.1.0 (BWA) [21] with the mem algorithm before converting to bam format and sorting with samtools v.1.13 [22]. PCR duplicates were removed with Picard MarkDuplicates [23], and mapping statistics were calculated with samtools *flagstat* and *idxstats*, bedtools *genomecov* [24], and mosdepth [25].

### 2.6. Determination of Karyotype from Sequence Data

To assess the sex chromosome complement, a number of measurements based on sequencing coverage ratios were calculated: the X:Y coverage ratio, X:autosome ratio, and Y:autosome ratio. As genetic females have two X chromosomes (XX) and males have one X and one Y (XY), a female karyotype would show roughly similar coverage of the X chromosomes compared to autosomes and low coverage of the Y chromosome. In a male (XY) karyotype, the single X chromosome would be expected to have approximately half the coverage of the autosomes, and also roughly half coverage for Y chromosome reads.

Chromosome 10 was selected as the comparative autosome.

### 2.7. SRY Promotor Mutation

Heidari et al. [15] identified a T to G mutation on the *SRY* promotor region of a sheep with Swyer syndrome, which they speculate may be implicated in the disease. To investigate whether the patient had this mutation, the Y chromosome region was extracted from bam alignment files and imported into Geneious v. 8.1.9 [26]. Consensus sequences were obtained specifying the fewest ambiguities option and coding bases with <3 read coverage coded as missing (N). The *SRY* promotor region corresponding to that used in the multi-species alignment of Heidari et al. [15] was extracted from each sample. Ten sequences from Heidari et al. [15] were downloaded from Genbank, representing their query sequence individual (*SRY*+, Swyer syndrome sheep), four male *Ovis* spp., two male *Capra hircus* samples, one male *Rupicapra pyrenaica* (chamois) sample, one male *Ammotragus lervia* (barbary) sample, and one male *Bos taurus* sequence. Alignments were performed with MUSCLE [27] in the Geneious platform [26], with default settings and the maximum number of iterations set to five. The final alignment was inspected for the T to G mutation at position 303 [15].

## 3. Results

### 3.1. Clinical Phenotype

On clinical examination, abnormalities were restricted to the reproductive organs. The patient presented with a mixed phenotype with female external genitalia (vulva with a markedly enlarged and prominent clitoris and vagina) (Figure 1a,b) and some male external characteristics such as a beard, larger body size, and larger horns (Figure 1c). Moreover, the patient expressed male-like behavior (mounting, aggression, and flehmen response).

### 3.2. Diagnostic Imaging

CT images revealed a fluid-filled tubular organ dorsal to the urinary bladder (Appendix A), which branched into a left and right horn to which bulbous soft tissue opacity gonads (Appendix A) were attached by narrow tethers.

### 3.3. Exploratory Laparotomy and Histological Examination

Exploratory laparotomy revealed the fluid-filled, bifurcated, hypoplastic uterus dorsal to the urinary bladder. The uterine horns terminated in white, thick-walled tissue, leading to what appeared grossly to resemble hypoplastic testicular gonads (Appendix A).

Histological assessment of the gonads confirmed paired testes encased in abundant adipose tissue (Figure 2). Bilaterally, the majority of the tissue was represented by well-developed epididymis, devoid of speratozoa. Each gonad contained a single nodule of seminiferous tubules formed by multiple lobular clusters of amphophilic polygonal cells, with large clear cytoplasmic vacuoles (presumptive Sertoli cells), with no evidence of spermatogenesis. Leydig cells were not visible in the interstitial tissue (Figure 3a). On one side, the vas deferens were accompanied by a uterine horn (Figure 3b).

### 3.4. Blood Testosterone Concentrations

The testosterone concentration prior to surgery was 2.10 nmol/L, while the result two weeks post-operatively was <0.03 nmol/L. This reduction is indicative of the presence of functional testicular tissue. Nevertheless, the pre-surgery testosterone value is low compared to the normal testosterone value for male white goats in January (7.31 nmol/L) [28].

### 3.5. Illumina Sequencing Read Mapping

DSD Goat had a similar X-fold sequencing coverage when aligned to either the San Clemente or Saanen reference genomes (15X, Appendix A). As the Saanen genome had a complete Y chromosome contig, this genome was used for further analysis. Downsampling of the comparative samples, Male Control and Female Control resulted in 16X and 15X coverage, respectively (Appendix A).

Read mapping ratios of Male Control and Female Control were consistent with what would be expected for a male XY and a female XX karyotype (Appendix A). The read mapping pattern of the Goat DSD suggested the presence of a Y chromosome (0.3M mapped reads), the coverage for the X chromosome was intermediate (12X) between the male (8X) and the female (14X), and median X coverage for the Y chromosome was much lower than in the male dairy goat—3X compared to 8X. DSD Goat also had a higher X:chr10 ratio (0.8) than the male (0.5) and a lower Y:chr10 ratio (0.2 compared to 0.5, Appendix A).

### 3.6. SRY Promotor Mutation

Read mapping of DSD Goat data to the *SRY* region showed lower read coverage than in the Male control. DSD Goat did not have this mutation, nor did it show any other mutation in the promotor region that may be associated with Swyer syndrome.

## 4. Discussion

This study describes a case of a horned pygmy goat presenting with a DSD. The majority of caprine DSD relates to the polled intersex syndrome; however, the presence of horns and the potential of being XY suggests that PIS is unlikely to be the underlying cause of this individual’s condition. Testicular feminisation syndrome or insensitivity to androgen hormones [29] is also unlikely as it prevents the formation of a uterus and secondary male characteristics. Rare instances of XY individuals with testicular tissue and female ductal systems derived from the embryonic Müllerian duct are classified as exhibiting persistent Müllerian duct syndrome (PMDS). This situation can arise due to failure to express functional *AMH* or its receptor, *AMHR2*. Various mutations in either *AMH* or *AMHR2* have been identified in humans [30], and it is well recognised in Miniature Schnauzer dogs, in which it is associated with an *AMHR2* mutation with an autosomal recessive mode of inheritance [31]. A single report is described in a goat, but no genetic basis was investigated [32].

The *SRY* gene is located on the Y chromosome and encodes the *SRY* protein, which is one of the *SOX9* transcription factors responsible for the development of the male reproductive tissues [33]. Deletions or mutations of *SRY* on any coding or regulatory region can disrupt male development and cause XY female sex reversal. In sheep, a mutation on the *SRY* promotor has been shown to lead to hermaphrodism [15], and, in cattle, deletion of *SRY* was associated with Swyer Syndrome-like condition in 7 out of 8 animals [11]. Humans with the condition lack the *SRY* gene (*SRY*-) in around 10–15% of cases, and mutations in the *SRY* gene (*SRY*+ individuals) are found in a further 10–15% of individuals [16,34]. Raudsepp et al. [35] suggest a distinction between *SRY-* and *SRY+* Swyer syndrome individuals in horses, where the former lacks testes, and the latter may exhibit underdeveloped testes.

Freemartinism, which describes XX female-to-male, intersex presentations when gestation is shared with a male twin, would be unlikely as this individual was reportedly a single kid. However, being single-born does not completely exclude this condition, as multiple single-born freemartin heifers have been reported [36]. This situation may occur as a result of in utero death of a male co-twin, which influenced the sexual development of its female twin (with varying degrees of masculinisation) prior to its loss. Another rare cause for an XX/XY karyotype is mosaicism, in which two or more populations of cells arise from a single zygote due to recombination or mutation in mitosis or meiosis [3].

Read mapping coverage to the Y chromosome and *SRY* regions is lower than is observed for the control male despite down-sampling data for even coverage across the genome. The chromosome coverage ratio for Y:chr10 was also lower than expected for a male. These findings align well with what would be expected from a mosaic or chimeric individual, where the presence of Y-chromosome-containing cells is reduced compared to a typical male, and X chromosome coverage is elevated relative to an XY male but lower than an XX female. However, pygmy goats are a composite breed with a complex history, including the merging of breeds with dwarfism and from disparate geographic origins. It would not be surprising if pygmy goats exhibited unique Y chromosome structure and genetic divergence from other goat breeds. However, they are missing from previous studies characterising the genome-wide [37] and Y-chromosome diversity [38] of goat breeds globally. Due to a dearth of available comparative data, it has not been possible to explore their Y chromosome phylogeny in detail here, although determining which major Y chromosome lineage pygmy goats belong to and whether Y chromosome introgression and admixture is observable would be informative.

Identification of chimera/mosaic specimens requires further cytogenetic testing, such as karyotyping. This involves cell culture, preparation, and staining [39] from a freshly collected blood sample. In this case, karyotyping was not possible due to the lack of funding at the time of the clinical case and the impossibility of collecting fresh blood samples after the discharge of the patient. If the opportunity arises to collect suitable blood and or tissue samples in the future, this analysis will be performed and reported as a follow-up study. New methods to perform cytological analysis on frozen blood that will help to facilitate this are becoming available, although the volumes of blood required can be prohibitive due to sample collection limitations from clinical cases [40]. Possible further work could also include PCR and Sanger sequencing of *SRY* and other male-specific Y-chromosomal genes (*DMRT1*, *USP9Y*, *UTY*, and *DDX3Y*, for example) and other key genes in sexual development, such as *AMHR2*. In addition, variant calling of this sample with the VarGoats dataset [38] and the generation of a pygmy goat breed-specific reference genome to avoid reference mapping bias would add valuable genomic resources for pygmy goats.

## 5. Conclusions

The pygmy goat, in this case, appears to be an XY DSD and is potentially *SRY*+. To confirm the karyotype, both conventional and molecular cytogenetic analysis will be required.

## Figures and Tables

**Figure 1 animals-15-00976-f001:**
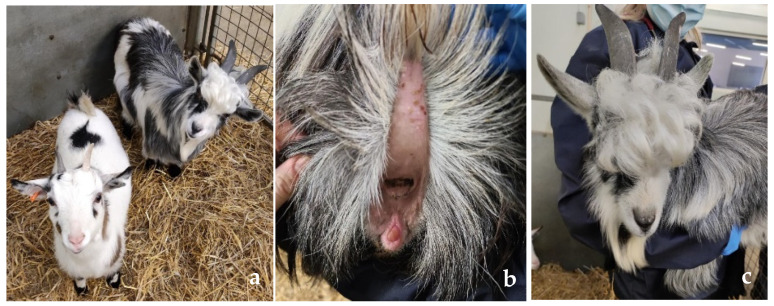
Phenotypical appearance of DSD goat. (**a**) Phenotypical differences between female pygmy goat (left) and DSD goat (right). Larger horns, beard, and hairier head are characteristics of the DSD goat; (**b**) Vulva with prominent clitoris. (**c**) Male-like phenotypic characteristics of the DSD goat.

**Figure 2 animals-15-00976-f002:**
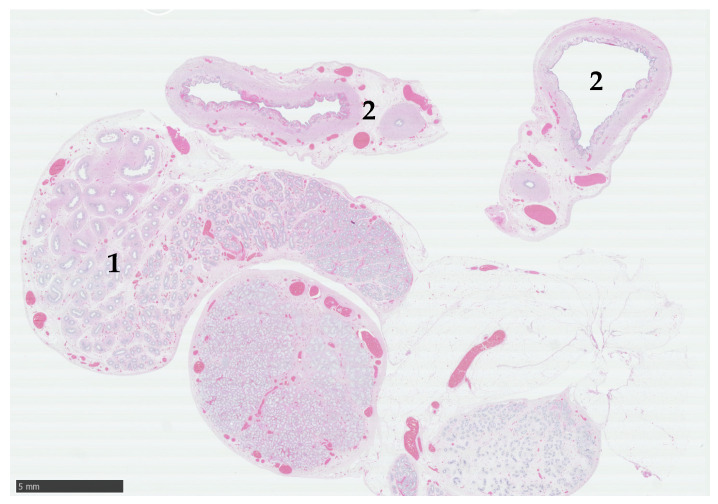
Histological section of gonad and tubular genitalia from a pygmy goat with a disorder of sexual development. The subgross section reveals a small testis with prominent epididymis (1), and two transverse sections of the associated tubular system show paired structures (2). Haematoxylin and eosin (H&E). Scale bar = 5 mm.

**Figure 3 animals-15-00976-f003:**
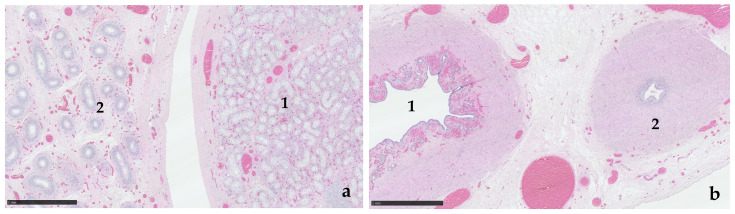
(**a**) Testicular tissue (1) displays numerous seminiferous tubules lined by vacuolated Sertoli cells. Efferent ductules within the epididymis (2) are devoid of spermatozoa. H&E. Scale bar = 1 mm. (**b**). Paired (unilateral) tubular genitalia. A well-vascularised glandular endometrium is present surrounding the larger-calibre tubular genital structure (1). A muscular ductus deferens is present (2). H&E. Scale bar = 1 mm.

## Data Availability

The raw whole genome sequencing data for the 10-month-old pygmy goat (DSD Goat) presented in the study are openly available in the Short Read Archive under BioProject PRJNA1232017.

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
