# Peer review of "Clinical, Histological and Genetic Characterisation of a Disorder of Sexual Development in a Pygmy Goat"

_animals, 2025, doi:10.3390/ani15070976_

Round 1
Reviewer 1 Report
Comments and Suggestions for Authors
The case report in question is very interesting as it deals with an individual with a horn, which is not reported in the literature, being common in polled individuals.
Examinations were carried out, including tomography, DNA extraction and sequencing, determination of karyotype, and SRY promoter mutation analysis. These are analyses not commonly performed in reports like this, making it a differential aspect of the paper. However, a karyotype analysis, both conventional and molecular cytogenetic, could have been conducted to confirm the XY individual.
Regarding the results, the structures present (epididymal ducts, layers of the uterus, for example) could be indicated with arrows in the images to facilitate understanding.
The conclusion needs improvement; it should clearly and objectively respond to the proposed objective.
Author Response
The case report in question is very interesting as it deals with an individual with a horn, which is not reported in the literature, being common in polled individuals.
Examinations were carried out, including tomography, DNA extraction and sequencing, determination of karyotype, and SRY promoter mutation analysis. These are analyses not commonly performed in reports like this, making it a differential aspect of the paper. However, a karyotype analysis, both conventional and molecular cytogenetic, could have been conducted to confirm the XY individual.
Response: We acknowledge the reviewer’s comment regarding the lack of cytogenetic analysis, which would indeed have provided valuable information about the underlying chromosomal abnormalities. Unfortunately, we were unable to perform these tests due to the limited availability of fresh blood samples and insufficient material, and funding for further analysis. Additionally, the genetic analysis was conducted several months after the clinical presentation, making it challenging to obtain the necessary samples for cytogenetic studies such as banding and FISH. Despite these constraints, we believe the whole-genome sequencing provided meaningful insights into the genetic landscape and contributed to understanding the case. We hypothesised that the read coverage would act as a proxy for the karyotype and that an XY or intermediate karyotype could be identified by the proportion of reads mapping to each sex chromosome.
Regarding the results, the structures present (epididymal ducts, layers of the uterus, for example) could be indicated with arrows in the images to facilitate understanding.
Response: Thank you for the suggestion. To improve clarity and facilitate understanding, we have added numbers in the figures corresponding to the identified structures, matching the description in the figure legends.
The conclusion needs improvement; it should clearly and objectively respond to the proposed objective.
Response: Thanks for your feedback. The conclusion has been revised and adjusted appropriately.
Reviewer 2 Report
Comments and Suggestions for Authors
The study of abnormalities in goats has a long history. Undoubtedly, each case merits attention, as it contributes to the broader understanding of the mechanisms underpinning normal development (gonadogenesis and gametogenesis) and possible abnormalities.
The main drawback of this study is the lack of cytogenetic analysis. It is useful to make a whole-genome sequencing, but the question of what the cause of the abnormalities was, the absence of the Y chromosome, its translocation or the translocation of its fragment containing the Sry gene, cannot be resolved without cytogenetic analysis, including bandings and FISH.
A second point of concern is the poor use of the whole genome obtained, with only a fragmentary analysis of the Sry gene. No attempt was made to analyse other genes, the PIS for example. It is not necessary that a mutation in a single gene alone would be the cause of an abnormal development, although this has been shown for humans (see McElreavey, K. and Bashamboo, A., 2023. Monogenic forms of DSD: an update. Hormone Research in Paediatrics, 96(2), pp.144-168). And it would be very useful to analyse at least the key genes (NROB1, Amh, and antiMüllerian hormone receptor 2 (AMHR2), DMRT1, WT1, etc.) for significant mutations.
There is no indication in the manuscript that the resulting genome or Sry gene will be deposited in public databases such as GenBank. This also decreases the value of the study.
The comparison with the phenomenon of freemartinism is not quite correct, because in this situation XX females suffer more due to earlier expression of genes of the male pathway, for example the Sry gene. The overall morphological and histological picture is more consistent with mosaicism. Fetal cells are known to persist in various maternal organs throughout life, and there is also the possibility that these cells may have an effect on subsequent embryos.
It is possible that this patient is a mosaic or XX karyotype with translocation of the Y fragment. It is unlikely that Zfy analysis will give a result because i) Zfy participates in spermatogenesis, not gonadogenesis, and ii) Zfy and Sry tend to be located close to each other and could be translocated together as a part of the Y chromosome.
In general, there are very few results in the manuscript, even though the whole genome of the animal was obtained. Without a karyotype, this patient is pointless to study and not particularly interesting even as a case study.
Typos. Lines 105, 118, 339, 356: 'karotype' instead of karyotype.
These papers are likely to be useful:
Paredes, J., Czochara, L., Villagomez, D. and King, A., 2024. Disorders of sexual development in small ruminants. Clinical Theriogenology, 16.
Montenegro, L., Costa, I., Maltez, L., Evaristo, V., Dias, I.R., Martins, C., Borges, I., Morinha, F., Pereira, R., Neto, N. and Oliveira, C., 2024. Unusual sex chromosomal DSD in a domestic Shorthair cat with a 37, X/38, XY mosaic karyotype. BMC Veterinary Research, 20(1), p.298.
Bianchi, D.W., Khosrotehrani, K., Way, S.S., MacKenzie, T.C., Bajema, I. and O’Donoghue, K., 2021. Forever connected: the lifelong biological consequences of fetomaternal and maternofetal microchimerism. Clinical chemistry, 67(2), pp.351-362.
Comitre-Mariano, B., Martínez-García, M., García-Gálvez, B., Paternina-Die, M., Desco, M., Carmona, S. and Gómez-Gaviro, M.V., 2022. Feto-maternal microchimerism: Memories from pregnancy. Iscience, 25(1).
Author Response
The study of abnormalities in goats has a long history. Undoubtedly, each case merits attention, as it contributes to the broader understanding of the mechanisms underpinning normal development (gonadogenesis and gametogenesis) and possible abnormalities.
The main drawback of this study is the lack of cytogenetic analysis. It is useful to make a whole-genome sequencing, but the question of what the cause of the abnormalities was, the absence of the Y chromosome, its translocation or the translocation of its fragment containing the Sry gene, cannot be resolved without cytogenetic analysis, including bandings and FISH.
Response: We acknowledge the reviewer’s comment regarding the lack of cytogenetic analysis, which would indeed have provided valuable information about the underlying chromosomal abnormalities. Unfortunately, we were unable to perform these tests due to the limited availability of fresh blood samples and insufficient material, and funding for further analysis. Additionally, the genetic analysis was conducted several months after the clinical presentation, making it challenging to obtain the necessary samples for cytogenetic studies such as banding and FISH. Despite these constraints, we believe the whole-genome sequencing provided meaningful insights into the genetic landscape and contributed to understanding the case. Our hypothesis was that the read coverage would act as a proxy for the karyotype and that an XY or intermediate karyotype could be identified by the proportion of reads mapping to each sex chromosome.
A second point of concern is the poor use of the whole genome obtained, with only a fragmentary analysis of the Sry gene. No attempt was made to analyse other genes, the PIS for example. It is not necessary that a mutation in a single gene alone would be the cause of an abnormal development, although this has been shown for humans (see McElreavey, K. and Bashamboo, A., 2023. Monogenic forms of DSD: an update. Hormone Research in Paediatrics, 96(2), pp.144-168). And it would be very useful to analyse at least the key genes (NROB1, Amh, and antiMüllerian hormone receptor 2 (AMHR2), DMRT1, WT1, etc.) for significant mutations.
Response: We appreciate the reviewer’s suggestion to conduct a broader analysis of key genes involved in sex development. Unfortunately, due to resources constraints and the retrospective nature of this study, our analysis focused primarily on the SRY gene, which was considered a priority based on the initial clinical presentation. We recognize the importance of investigating other critical genes, and we agree that a more comprehensive genetic analysis could provide further insights. We hope to pursue this in future studies, should additional resources become available.
There is no indication in the manuscript that the resulting genome or Sry gene will be deposited in public databases such as GenBank. This also decreases the value of the study.
Response: We appreciate the reviewer’s comment regarding data availability. At the time of submission, the data was in the process of being uploaded and was not yet publicly accessible. However, the data has now been uploaded to the Sequence Read Archive (SRA) under BioProject PRJNA1232017, ensuring public access and enhancing the transparency and value of our study.
The comparison with the phenomenon of freemartinism is not quite correct, because in this situation XX females suffer more due to earlier expression of genes of the male pathway, for example the Sry gene. The overall morphological and histological picture is more consistent with mosaicism. Fetal cells are known to persist in various maternal organs throughout life, and there is also the possibility that these cells may have an effect on subsequent embryos.
It is possible that this patient is a mosaic or XX karyotype with translocation of the Y fragment. It is unlikely that Zfy analysis will give a result because i) Zfy participates in spermatogenesis, not gonadogenesis, and ii) Zfy and Sry tend to be located close to each other and could be translocated together as a part of the Y chromosome.
In general, there are very few results in the manuscript, even though the whole genome of the animal was obtained. Without a karyotype, this patient is pointless to study and not particularly interesting even as a case study.
Response: We appreciate the reviewer’s detailed feedback and acknowledge the complexities surrounding the interpretation of this case. Regarding the comparison with freemartinism, our intention was to draw parallels with certain aspects of abnormal sexual development rather than suggest a direct correlation. We agree that mosaicism or an XX karyotype with translocation of a Y fragment are plausible explanations, and further cytogenetic analysis would be necessary to clarify this. We have expanded this on the discussion (lines 263-266 / 275-279). Unfortunately, as above mentioned, karyotyping and other cytogenetic methods could not be performed.
As for the Zfy analysis, we acknowledge the reviewer’s point that Zfy plays a role in spermatogenesis rather than gonadogenesis and this has been removed from the discussion.
Regarding the dataset and study scope, we recognize that the absence of cytogenetic data limits the interpretation of the findings. Nevertheless, we believe the whole-genome sequencing offers valuable insights into the genetic landscape and represents a starting point for further investigation into disorders of sexual development. We hope that making the genome publicly accessible (SRA BioProject ID PRJNA1232017) will enable future comparative analyses and contribute to the broader understanding of such cases.
We appreciate the constructive feedback and have revised the manuscript to reflect these limitations more explicitly.
Regarding the references provided, they have been reviewed and they have been useful for the review process and revision of the manuscript. We thank you for pointing out these typos. These have been addressed and changed.
Reviewer 3 Report
Comments and Suggestions for Authors
The manuscript entitled ”Clinical, histological and genetic characterization of a disorder of sexual development in pygmy goat” submitted by Castro et al. describes a case study of a disorder of sexual development in pygmy goat.
One individual goat with both female and male characteristics was subjected to clinical examination, diagnostic imaging analysis, surgical and histological examination, and genetic analysis by whole genome sequencing. An interesting study is presented, which clearly contributes to the understanding of disorder of sexual development. However, I find the results from the study somewhat preliminary as several questions are still open for further studies - and should be addressed. The results on the karyotype, of the goat presenting an XY male karyotype seems not completely conclusive. Similarly, the genetic and epigenetic studies of the SRY gene are very limited and should be extended. For instance, the DNA methylation profile of the SRY gene including the promoter could be studied in the patient and compared to the controls. Another example is the examination of blood testosterone concentrations. Only two values for the patient is presented and not commented or discussed. What is the normal range of testosterone concentration in pygmy goats? In addition, I believe that the Discussion section would benefit with some reorganizing and strengthening of the results obtained in the study. In conclusion, I recommend revision of the manuscript.
Minor points:
Keywords should be ordered alphabetically.
The position of the SRY promoter T/G mutation should be specified in relation to the transcription start site (TSS).
Figure 1: Add a, b, and c to the individual panels.
Figure 2: Add a and b to the two panels.
Line 203: What does Figure A3 mean? Should be Fig. 3?
Figure 3: What is 216?
Figure 4: Scale bar is lacking in the figure.
Figure 5: Scale bars are lacking.
Legend to Table 1. Goat 6 and 8 are not mentioned in the M & M section Animal Material.
Line 286: Heidari et al. [20].
Author Response
The manuscript entitled ”Clinical, histological and genetic characterization of a disorder of sexual development in pygmy goat” submitted by Castro et al. describes a case study of a disorder of sexual development in pygmy goat.
One individual goat with both female and male characteristics was subjected to clinical examination, diagnostic imaging analysis, surgical and histological examination, and genetic analysis by whole genome sequencing. An interesting study is presented, which clearly contributes to the understanding of disorder of sexual development. However, I find the results from the study somewhat preliminary as several questions are still open for further studies - and should be addressed.
Response: We appreciate the reviewer’s acknowledgment of our study and its contribution to understanding disorders of sexual development. We recognize that some questions remain open, and we acknowledge the preliminary nature of the findings due to certain limitations, such as the lack of cytogenetic analysis and a broader investigation of key genes beyond SRY. These aspects would indeed provide a more comprehensive understanding and are valuable directions for future research. For these reasons, this work has been submitted as a communication rather than a research article.
We have revised the manuscript to highlight these limitations and propose further studies to address them (lines 275-279 / 283-286). Thank you for your thoughtful feedback.
The results on the karyotype, of the goat presenting an XY male karyotype seems not completely conclusive. Similarly, the genetic and epigenetic studies of the SRY gene are very limited and should be extended. For instance, the DNA methylation profile of the SRY gene including the promoter could be studied in the patient and compared to the controls.
Response: We are grateful to the reviewer for highlighting these important points regarding the karyotype and the limited scope of the genetic and epigenetic studies of the SRY gene. Unfortunately, due to the retrospective nature of this study and the lack of funding and suitable samples, we were unable to conduct further cytogenetic analyses or explore epigenetic modifications such as DNA methylation profiling of the SRY gene. We recognize that such investigations, particularly comparing the patient to appropriate controls, could provide deeper insights into the gene’s regulation and its role in this case of sexual development disorder. We have revised the manuscript to acknowledge these limitations and propose these analyses as valuable directions for future research (lines 275-279 / 283-286).
Another example is the examination of blood testosterone concentrations. Only two values for the patient is presented and not commented or discussed. What is the normal range of testosterone concentration in pygmy goats?
Response: "Thank you for your feedback. We have added a sentence comparing the patient’s testosterone values to the normal range for male goats based on Polat et al. (2011), providing context for interpretation (lines 205-208). Please note that we could not find current literature on testosterone values on pigmy goats, so it has been extrapolated from other work in goats. It is worth noting that there is variation between different published work and that testosterone values also vary throughout the year”.
In addition, I believe that the Discussion section would benefit with some reorganizing and strengthening of the results obtained in the study. In conclusion, I recommend revision of the manuscript.
Response: Thank you for your constructive feedback. We have carefully revised and reorganized this section to provide a clearer interpretation of the findings and to place them in a broader context of disorders of sexual development. We believe these changes enhance the clarity and impact of the discussion.
Minor points:
Keywords should be ordered alphabetically.
The position of the SRY promoter T/G mutation should be specified in relation to the transcription start site (TSS).
Figure 1: Add a, b, and c to the individual panels.
Figure 2: Add a and b to the two panels.
Line 203: What does Figure A3 mean? Should be Fig. 3?
Figure 3: What is 216?
Figure 4: Scale bar is lacking in the figure.
Figure 5: Scale bars are lacking.
Legend to Table 1. Goat 6 and 8 are not mentioned in the M & M section Animal Material.
Line 286: Heidari et al. [20].
Response: We thank you for pointing out all these typos and minor errors. These have been addressed and changed.
Round 2
Reviewer 2 Report
Comments and Suggestions for Authors
Although the authors did not add any new material or bioinformatic data, they did improve the article as a whole.
Reviewer 3 Report
Comments and Suggestions for Authors
The forwarded revised manuscript has been significantly improved. I am satisfied with all answers , comments and corrections. I recommend acceptance.